# THE FOUNDATIONS OF TOKENIZATION: STATISTICAL AND COMPUTATIONAL CONCERNS

**Juan Luis Gastaldi**[1]     **John Terilla**[2]
**Luca Malagutti**[1]     **Brian DuSell**[1]     **Tim Vieira**[1]     **Ryan Cotterell**[1]
[1]ETH Zürich     [2]City University of New York
{juan.luis.gastaldi,lmalagutti,brian.dusell,ryan.cotterell}@inf.ethz.ch
jterilla@gc.cuny.edu   tim.f.vieira@gmail.com

## ABSTRACT

Tokenization—the practice of converting strings of characters from an alphabet into sequences of tokens over a vocabulary—is a critical step in the NLP pipeline. The use of token representations is widely credited with increased model performance but is also the source of many undesirable behaviors, such as spurious ambiguity or inconsistency. Despite its recognized importance as a standard representation method in NLP, the theoretical underpinnings of tokenization are not yet fully understood. In particular, the impact of tokenization on language model estimation has been investigated primarily through empirical means. The present paper contributes to addressing this theoretical gap by proposing a unified formal framework for representing and analyzing tokenizer models. Based on the category of stochastic maps, this framework enables us to establish general conditions for a principled use of tokenizers and, most importantly, the necessary and sufficient conditions for a tokenizer model to preserve the consistency of statistical estimators. In addition, we discuss statistical and computational concerns crucial for designing and implementing tokenizer models, such as inconsistency, ambiguity, finiteness, and sequentiality. The framework and results advanced in this paper contribute to building robust theoretical foundations for representations in neural language modeling that can inform future theoretical and empirical research.

## 1   INTRODUCTION

As a critical step in the natural language processing (NLP) pipeline, tokenization generally refers to the process of breaking up sequences of symbols into subsequences that can be represented as units, known as tokens. The tokenization of linguistic data has long been a common practice in the processing of natural language (cf. Palmer, 2000; Jurafsky & Martin, 2024). However, the significance of tokenizers took a turn with the emergence of deep neural models for NLP, where the representation of linguistic units plays a renewed fundamental role. With the development and widespread adoption of the *byte-pair encoding* (BPE) algorithm (Sennrich et al., 2016), subword tokenization became the privileged representation method for neural NLP. Adapting an existing compression algorithm (Gage, 1994) to overcome the challenges of out-of-vocabulary (OOV) terms in the context of neural machine translation, BPE quickly replaced previous heuristic and rule-based tokenizer models such as Morfessor (Creutz & Lagus, 2002) and Moses (Koehn et al., 2007), and was soon followed by other data-driven models, including *WordPiece* (Wu et al., 2016, following Schuster & Nakajima, 2012) and *Unigram* (Kudo, 2018) among the most widely adopted (cf. Mielke et al., 2021, for a survey).

The importance of subword tokenization for language models (LMs) has grown ever since, and tokenization methods are now built into standard language modeling toolkits, remaining the only major step not fully integrated into widely used end-to-end neural models. Among their recognized benefits, two are often advanced in the literature. Tokenizers offer the ability to train language models over an open vocabulary, circumventing the difficulties associated with OOV terms (Sennrich et al., 2016). In addition, tokenization is often described as an efficient, lossless *encoding* of the original data (Zouhar et al., 2023a). Moreover, based on empirical evidence of different kinds, tokenization has been hypothesized to introduce a helpful inductive bias in language modeling (Nawrot et al., 2023; Schmidt et al., 2024; Uzan et al., 2024), although in the current state of the art, this hypothesis

remains an open question. At the same time, tokenizers have also been in the spotlight for exhibiting undesirable behaviors that can have a negative impact on LMs. To name just a few, tokenization can be the source of spurious ambiguity (Kudo, 2018; Cao & Rimell, 2021), generate alignment issues (Poesia et al., 2022; Athiwaratkun et al., 2024), hinder robustness (Kudo, 2018; Xue et al., 2022), neglect relevant linguistic features (Bostrom & Durrett, 2020; Hofmann et al., 2021; Gow-Smith et al., 2022; Beinborn & Pinter, 2023) or result in inconsistent scoring in the use of LMs in other scientific fields, like psycholinguistics (Salazar et al., 2020; Kauf & Ivanova, 2023; Giulianelli et al., 2024).

The prominence of undesirable behaviors induced by tokenization, together with the lack of conclusive theoretical explanations for either their positive or negative effects in language modeling, motivated several recent attempts to dispense with tokenization altogether (Xue et al., 2022; Clark et al., 2022; Wang et al., 2024, *inter alia*). However, in the current state of research, the practical benefits of token representations in neural language modeling seem to outweigh their disadvantages, indicating that there is something to be understood rather than discarded in the process of tokenization.

The study of tokenization models has been an active area of research in recent years. Most of the work in this direction has been driven by an empirical perspective (Ding et al., 2019; Hou et al., 2023; Domingo et al., 2023; Fujii et al., 2023, *inter alia*). However, some notable exceptions exist where the authors have adopted a more theoretical approach (Guo, 1997; Kudo, 2018; Zouhar et al., 2023b;a; Berglund & van der Merwe, 2023; Rajaraman et al., 2024). The above-cited contributions notwithstanding, this paper contends there is still a need for a more foundational perspective. Among others, such a perspective should provide the means to advance results on the impact of tokenization on language model estimation, which are conspicuously absent from the literature. A foundational approach should also contribute to analyzing known issues in tokenization in a formal way, ultimately informing future theoretical and empirical research and contributing to increasing the reliance on models in situations in which properties such as formal guarantees, verification, or interpretability are as important as performance for an LM.

Accordingly, the objective of the present paper is to take a step forward toward a robust theoretical grounding for neural NLP by laying the foundations of tokenization from a formal perspective. To that end, we characterize the problem of tokenization in current language modeling as arising from the fact that, in practice, starting from an alphabet $\Sigma$ of elementary units, one seeks to estimate a probability distribution over $\Sigma^*$ indirectly, that is, by estimating a probability distribution over sequences of tokens in $\Delta^*$, where the set of tokens $\Delta$ is, in general, different from $\Sigma$. Therefore, the problem of tokenization is determined by the forward and backward mappings between $\Sigma^*$ and $\Delta^*$. To address this problem, we propose a formal framework based on what we found to be the simplest mathematical tool allowing us to characterize tokenizer models in their full generality, namely the category of stochastic maps. The proposed framework enables us to establish general conditions for a principled use of tokenizers. Crucially, we prove the necessary and sufficient conditions for a tokenizer model to preserve the consistency of statistical estimation of language modeling from data. Additionally, this paper aims to advance the theoretical understanding of existing challenges associated with tokenization, particularly those pertaining to inconsistency, ambiguity, finiteness, and sequentiality. To achieve this, we characterize these known issues through the lens of formal properties of composable maps, such as injectivity, multiplicativity, and bounded variation.

The outline of the paper is as follows. In §2, we present preliminary notions, including elementary aspects of formal language theory and the concept of stochastic maps, extending some existing results to cover the case of countably infinite sets. We also provide notational and terminological remarks. In §3, we propose a unified formal framework for representing and analyzing tokenizer models and establish various results for their use, including the necessary and sufficient conditions for a tokenizer model to preserve the consistency of estimators. Finally, in §§ 4 and 5 we discuss, from a formal perspective, statistical and computational concerns relevant to the study, design, and implementation of tokenizer models.

## 2 PRELIMINARIES

### 2.1 FORMAL LANGUAGES, ESTIMATORS, AND STOCHASTIC MAPS

An **alphabet** $\Gamma$ is a finite, nonempty set of **symbols**. The set $\Gamma^n$ consists of **strings** of symbols of length $n$. The symbol $\varepsilon$ denotes the empty string of length 0. The union $\Gamma^* \stackrel{\text{def}}{=} \bigcup_{n=0}^{\infty} \Gamma^n$ consists of

all finite strings (including $\varepsilon$) from the alphabet $\Gamma$. Similarly, we denote by $\Gamma^{\leq N}$ the set of all strings from $\Gamma$ of length less than or equal to $N$.

String concatenation ($\cdot$) is an associative product $\Gamma^* \times \Gamma^* \dot{\to} \Gamma^*$ for which $\varepsilon$ is an identity element. The triple $(\Gamma^*, \cdot, \varepsilon)$ defines a **monoid**, which is, in fact, a model of the free monoid on the set $\Gamma$. A **language** $L$ over an alphabet $\Gamma$ is a set of strings $L \subseteq \Gamma^*$. A **language model** $p$ is a probability distribution over $\Gamma^*$, i.e., $p$ is a function $p : \Gamma^* \to [0, 1]$ such that $\sum_{\gamma \in \Gamma^*} p(\gamma) = 1$. Language models generalize languages in the sense that the support of a language model, i.e., $\text{supp}(p) = \{\gamma \mid p(\gamma) \neq 0\}$, is a language. The definition of a language model as a probability distribution on $\Gamma^*$ is deliberately broad. In particular, note that no compatibility between $p$ and the monoidal structure in $\Gamma^*$ is assumed.[1]

In NLP, practitioners generally seek to **estimate** a language model $p$ from examples of naturally occurring text. Formally, the modeler assumes there exists a true distribution $p^\star$ over $\Gamma^*$, and considers a multiset of naturally occurring texts $\{\gamma_m\}_{m=1}^M \subset \Gamma^*$ to be samples from $p^\star$. In its most general form, an estimator of $p^\star$ is a sequence $\{p_n\}$ of probability distributions on $\Gamma^*$ such that $p_n$ becomes closer to $p^\star$ as $n$ increases. We call an estimator **consistent** if the sequence $\{p_n\}$ converges *pointwise* to $p^\star$.[2] More precisely, given a probability distribution $p^\star \colon \Gamma^* \to [0, 1]$ , and a sequence of distributions $\{p_n \colon \Gamma^* \to [0, 1]\}$, we say that $\{p_n\}$ is a consistent estimator of $p^\star$ if and only if, for all strings $\gamma \in \Gamma^*$, the sequence of numbers $\{p_n(\gamma)\}$ converges to the number $p^\star(\gamma)$.

This notion of consistent estimation is general enough to include many estimation methods, where the $p_i$ (with $1 \leq i \leq n$) can depend on various properties of the sample, such as the size $M$, and may be parameterized by a set of parameters $\theta$. In particular, the relationship between the data $\{\gamma_m\}$ and the estimator $\{p_n\}$ is determined by the practitioner through the choice of an estimation method, e.g., **maximum likelihood estimation** (MLE; Lehmann & Casella, 1998). MLE is often used in language model estimation where it corresponds to determining $\{p_n\}$ by minimizing the **cross entropy loss** on the data. As such, MLE amounts to minimizing the **relative entropy**, also called **the Kullback–Leibler divergence**, $D_{\mathrm{KL}}(p^\star \parallel p_n)$ between $p^\star$ and $p_n$, making $\{p_n\}$ a consistent estimator of $p^\star$. Note that this requires a stronger form of convergence than the one we use in our framework: if $D_{\mathrm{KL}}(p^\star \parallel p_n) \to 0$ then $p_n \to p^\star$ pointwise (a consequence of Pinsker's lemma). By adopting a weak kind of convergence, our definition is, therefore, compatible with a wide variety of convergence measures while remaining relatively easy to check for.

Our definition of tokenizer models will require the use of a special kind of map between sets called a stochastic map. The reference Baez & Fritz (2014) contains a detailed introduction to the category of finite sets with stochastic maps between them. Here, we will extend some of the results in Baez & Fritz (2014) to cover the case of countably infinite sets. We assume all sets are countable, either finite or countably infinite. A **stochastic map** from a set $X$ to a set $Y$ is a function from $X$ to the set of probability distributions on $Y$. We use

$$X \rightsquigarrow Y$$

to denote a stochastic map from $X$ to $Y$ and the notation $x \mapsto f(y \mid x)$ to denote the probability of $y \in Y$ in the distribution assigned to $x \in X$. In other words, a stochastic map $f \colon X \rightsquigarrow Y$ is a function

$$X \times Y \to [0, 1]$$
$$(x, y) \mapsto f(y \mid x)$$

satisfying $\sum_{y \in Y} f(y \mid x) = 1$ for all $x \in X$. The notation $f(y \mid x)$ is evocative of the conditional probability of $y$ given $x$, but it is more accurate to think of *indexing* by $x$ rather than *conditioning* on $x$ because there is no assumption that the numbers $f(y \mid x)$ can be assembled into a joint distribution on $X \times Y$.

Significantly, stochastic maps can be composed. The composition

$$X \overset{f}{\rightsquigarrow} Y \overset{g}{\rightsquigarrow} Z$$
$$\underset{gf}{\underbrace{\rightsquigarrow}}$$

$gf \colon X \rightsquigarrow Z$ is defined by

---

[1] In addition, one could typically require, for instance, that $p(\gamma \cdot \gamma') \leq \min\{p(\gamma), p(\gamma')\}$.

[2] Following the usual convention, we may denote convergence as $\{p_n\} \to p^\star$, which is not to be confused with the notation for functional types (e.g., $\Sigma^* \to \Delta^*$). The context of use should prevent any ambiguity.

$$gf(z \mid x) = \sum_{y \in Y} g(z \mid y)f(y \mid x). \tag{1}$$

Since the sum in equation 1 contains infinitely many summands, it requires to check that the formula for $gf(z \mid x)$ is finite, and that for each $x \in X$, $gf(\cdot \mid x)$ defines a probability distribution on $Z$, both of which follow from the fact that:

$$\sum_{z \in Z} gf(z \mid x) = \sum_{z \in Z} \sum_{y \in Y} g(z \mid y)f(y \mid x) = \sum_{y \in Y} \sum_{z \in Z} g(z \mid y)f(y \mid x) = \sum_{y \in Y} f(y \mid x) = 1.$$

If one arranges a stochastic map into an $|X| \times |Y|$ matrix with the $f(y \mid x)$ entry in the $x, y$ position, then every entry is nonnegative and the sum of every row is 1. The computation above shows that composition of stochastic maps is realized by matrix multiplication, and that—even when the matrices are infinite—the row-column dot products are finite, and the result of matrix multiplication is again a matrix with nonnegative entries whose rows sum to 1. This view makes it clear that composition of stochastic maps is associative.

Stochastic maps generalize both ordinary probability distributions and functions. A probability distribution over a set $X$ can be represented as a stochastic map into $X$ from a 1-element set, denoted as $\mathbf{1} \stackrel{\text{def}}{=} \{1\}$, i.e., $p \colon \mathbf{1} \rightsquigarrow X$. In such cases, the customary notation $p(x)$ can be used without risk of ambiguity as a shorthand for the more cumbersome $p(x \mid 1)$. An ordinary function $f \colon X \to Y$ can be regarded as a stochastic map $X \rightsquigarrow Y$ by mapping $x$ to the probability distribution on $Y$ concentrated on the singleton $\{f(x)\}$, in which case we say the stochastic map $f$ is **deterministic**. For simplicity, when a stochastic map $f \colon X \rightsquigarrow Y$ is deterministic, writing $y = f(x)$ means that $f(y \mid x) = 1$ and $f(y' \mid x) = 0$ for $y' \neq y$. Composition generalizes both the composition of functions and the pushforward of a probability function via a function. If $p \colon \mathbf{1} \rightsquigarrow X$ is a probability distribution on $X$ and $f \colon X \to Y$ is a deterministic function, then the composition

$\mathbf{1} \stackrel{p}{\rightsquigarrow} X \stackrel{f}{\rightsquigarrow} Y$ is a stochastic map $fp \colon \mathbf{1} \rightsquigarrow Y$, which is a probability distribution on $Y$ whose formula is $fp(y) = \sum_{x \in X} f(y \mid x)p(x \mid 1) = \sum_{x \in f^{-1}(y)} p(x)$. That is, $fp$ is just the pushforward of the probability distribution $p$ via the function $f$.

For any set $X$, the identity function on $X$ behaves as an identity for stochastic maps. That is $\mathrm{id}_X \colon X \rightsquigarrow X$ is the stochastic map defined by $\mathrm{id}_X(x' \mid x) = 1$ when $x' = x$ and $\mathrm{id}_X(x' \mid x) = 0$ when $x' \neq x$. In matrix representation, $\mathrm{id}_X$ is the identity matrix, and satisfies $f\,\mathrm{id}_X = f = \mathrm{id}_Y f$ for all stochastic maps $f \colon X \rightsquigarrow Y$. Stochastic maps also come equipped with natural notions of injectivity and surjectivity. The **support** of the stochastic map $f$ is the union of the support of the distributions $f(\cdot \mid x)$ as $x$ ranges over $X$. A stochastic map $f \colon X \rightsquigarrow Y$ is **injective** iff the supports of $f(\cdot \mid x)$ and $f(\cdot \mid x')$ are disjoint whenever $x \neq x'$. A stochastic map $f \colon X \rightsquigarrow Y$ is **surjective** iff, for all $y \in Y$, there exists $x \in X$ such that $f(y \mid x) \neq 0$. Injectivity and surjectivity for stochastic maps reduce to their ordinary definitions in the case of deterministic maps.

## 2.2 Notation and Terminology

We adopt the following notational conventions. We denote the **length** $n$ of a string $\boldsymbol{\gamma} \in \Gamma^n$ as $|\boldsymbol{\gamma}|$. The expression $\boldsymbol{\gamma}' \preceq \boldsymbol{\gamma}$ denotes the fact that $\boldsymbol{\gamma} = \boldsymbol{\gamma}' \cdot \boldsymbol{\gamma}''$ for $\boldsymbol{\gamma}, \boldsymbol{\gamma}', \boldsymbol{\gamma}'' \in \Gamma^*$, that is, $\boldsymbol{\gamma}'$ is a **prefix** of $\boldsymbol{\gamma}$. Alphabets will be denoted by uppercase Greek letters (e.g., $\Gamma$, B). In the context of tokenization, we will be interested in maps between strings of languages over two different alphabets, which we will denote $\Sigma$ and $\Delta$. For a more intuitive presentation that avoids ambiguity, we reserve the term **alphabet** for the former and call the latter **vocabulary**, systematically using the green and violet colors to highlight the respective relation to either of these sets. We denote symbols by lowercase Greek letters, e.g., $\sigma \in \Sigma$ and $\delta \in \Delta$, calling them **characters** in the first case and **tokens** in the second. Strings will be denoted by bold lowercase Greek letters, e.g., $\boldsymbol{\sigma} \in \Sigma^*$ and $\boldsymbol{\delta} \in \Delta^*$, reserving the name character strings or **texts** for the former and token strings or **token sequences** for the latter. The reader should keep in mind these terminological distinctions are for expository purposes only. From the formal perspective advanced in this paper, we do not assume any inherent privilege of $\Sigma$ over $\Delta$, focusing instead on how their respective elements can be mapped into each other. Maps are denoted using the color of their codomains.

When necessary, we will distinguish the empty character string $\varepsilon_\Sigma \in \Sigma^*$ from the empty token sequence $\varepsilon_\Delta \in \Delta^*$. Examples of strings and tokens will be written in monospace font (e.g., t, the). There are cases where $\Delta \cap \Sigma^* \neq \emptyset$, and it will be necessary to distinguish between concatenation in $\Sigma^*$ and $\Delta^*$. In $\Delta^*$, concatenation will be denoted as ⌐. So, for example, if $\Sigma = \{t, h, e\}$ and $\Delta = \{th, he, e\}$, the expression t·h·e denotes the concatenation in $\Sigma^*$ of the three characters t, h, and e, while the expression th⌐he represents the concatenation in $\Delta^*$ of the two tokens t and he. The cases when $\Delta \cap \Sigma^* \neq \emptyset$ are of sufficient significance that we shall generally avoid using the simple juxtaposition of characters to express concatenation. Therefore, the reader should always interpret th as a token in $\Delta$, and not a text in $\Sigma^*$ (written t·h). If further notational clarification is needed, square brackets may be used to represent the concatenation of two texts in $\Sigma^*$ (and likewise for $\Delta^*$). For example, $[\text{t·h}]$·e denotes the concatenation of the text t·h with the character e in $\Sigma^*$. Should any ambiguity arise between specific characters and tokens (e.g., $t \in \Sigma$ vs. $t \in \Delta$), it will be explicitly disambiguated whenever there is a risk that context alone is insufficient.

## 3 A FORMAL FRAMEWORK FOR TOKENIZATION

As observed in the previous pages, in modern NLP, the problem of tokenization arises from the fact that one seeks to estimate a model $p^\star$ over strings of symbols in one alphabet *indirectly*, that is, by estimating a probability distribution $q$ over strings of symbols on a different alphabet. Therefore, from a strictly formal perspective, the problem of tokenization can be characterized as that of the respective mappings between two sets of strings, conventionally referred to as the set $\Sigma^*$ of character strings and the set $\Delta^*$ of token sequences. In order to estimate $p^\star$ through $q$, $\Sigma^*$ needs to be mapped into and from $\Delta^*$. The connection between $\Sigma^*$ and $\Delta^*$ is thus made through a pair of mappings $(\tau, \kappa)$ that constitutes the basis of our formal characterization of tokenization. Accordingly, in its most general form, a tokenizer can be defined as follows:

**Definition 3.1.** *A **tokenizer model** (or simply **tokenizer**) from $\Sigma^*$ to $\Delta^*$ is a pair of stochastic maps $\mathcal{T} = (\tau, \kappa)$, respectively called the **encoder** and the **decoder**, where the encoder is a stochastic map $\tau \colon \Sigma^* \rightsquigarrow \Delta^*$, and the decoder is a stochastic map $\kappa \colon \Delta^* \rightsquigarrow \Sigma^*$.*

Definition 3.1 is deliberately broad, covering *any* pair of string-to-string mappings $\tau$ and $\kappa$. Other than the fact that the domain of each mapping constitutes the codomain of the other, we define the encoder and decoder as arbitrary stochastic maps. In other words, we will be regarding $\tau$ and $\kappa$ primarily from the point of view of their *composition*. In particular, we do not require any specific connection between the alphabet $\Sigma$ and the vocabulary $\Delta$, and hence the use of the terms encoder and decoder is also strictly conventional. However, the distinction is motivated by an implicit assumption behind the established use of tokenizers in language models—namely, that the samples $\{\sigma_m\}_{m=1}^M \subset \Sigma^*$ of naturally occurring texts used for estimation can be mapped into $\Delta^*$ in such a way that the estimated model $q$ can be, in turn, transformed into a model $p$ over $\Sigma^*$ through the map $\kappa$, such that $p = \kappa q$ can be considered as an estimate of the original distribution $p^\star$. When $p$ is given in the form of $\kappa q$, we say that $p$ is a **tokenized language model**.

Despite the potential empirical increase in the predictive performance of a model resulting from specific tokenization choices, the soundness of such a procedure is not guaranteed for arbitrary $\tau$ and $\kappa$ without further conditions. On the one hand, the notion of estimation in $\Delta^*$ is not well defined unless there exists a reference distribution $q^\star$ over $\Delta^*$ to which the estimator $\{q_n\}$ can converge. On the other, assuming such an estimator is consistent, transforming it into a consistent estimator of $p^\star$ requires a way to map the sequence $\{q_n\}$ into a sequence $\{p_n\}$ that converges to $p^\star$.

Assuming a reference distribution $p^\star$ exists on $\Sigma^*$, one obtains a reference $q^\star$ on $\Delta^*$ simply through the composition (Eq. (1)) with the encoder: $q^\star = \tau p^\star$. In other words, the following diagram of stochastic maps commutes

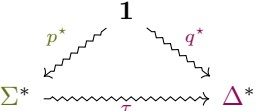

The distribution $q^\star$ is just the **pushforward** of the measure $p^\star$ along $\tau$, which then makes the encoder $\tau$ a *measure-preserving map* between $(\Sigma^*, p^\star)$ and $(\Delta^*, q^\star)$.

In the same way, $\{p_n\}$ can be obtained by mapping the sequence $\{q_n\}$ through $\kappa$. Defining $p_i = \kappa q_i$, we obtain the following commutative diagram

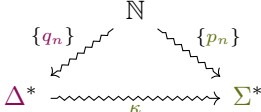

So far, none of these requirements imposes conditions on $\tau$ and $\kappa$ other than being well-defined mappings between their respective domains and codomains. Notably, the notion of estimation of $\tau p^\star$ is well defined for arbitrary $\tau$. However, given a consistent estimator $\{q_n\}$ of $q^\star$, $\{\kappa q_n\}$ *is not guaranteed to converge to* $p^\star$ without further conditions on $\kappa$. To establish such conditions, we will need the following lemmas.[3]

**Lemma 3.1.** *Let $\{p_n\}$ be a sequence of probability distributions over a countable set $X$ that converges pointwise to a probability distribution $p$. Then $\lim_{n\to\infty} \sum_{x\in X} |p_n(x) - p(x)| = 0$.*

**Corollary 3.0.1.** *Let $\{p_n\}$ be a sequence of probability distributions over a countable set $X$ that converges pointwise to a probability distribution $p$. Then $\{p_n\} \to p$ uniformly.*

**Lemma 3.2.** *Let $f$ be a stochastic map from $X$ to $Y$, and $\{p_n\}$ be an estimator for a probability distribution $p$ on $X$. Then $fp_n$ is an estimator for the probability distribution $fp$ on $Y$.*

In other words, Lemma 3.2 says that stochastic maps preserve the consistency of estimators. Armed with Lemma 3.2, it is now easy to establish a simple but *fundamental* principle for the use of tokenization models in language modeling.

**Theorem 3.1** (Fundamental Principle of Tokenization)**.** *Given a reference probability distribution $p^\star$ over $\Sigma^*$, a tokenizer $\mathcal{T} = (\tau, \kappa)$ from $\Sigma^*$ to $\Delta^*$, and a consistent estimator $\{q_n\}$ of the image reference distribution $q^\star = \tau p^\star$, the sequence $\{\kappa q_n\}$ is a consistent estimator of $p^\star$ if and only if $\kappa \tau p^\star = p^\star$.*

Theorem 3.1 characterizes precisely when a consistent estimator $\{q_n\}$ of $q^\star$ yields a consistent estimator $\{p_n\}$ of $p^\star$ after decoding. Based on the fundamental principle expressed in Theorem 3.1, we propose the following definitions:

**Definition 3.2.** *Given a probability distribution $p$ over $\Sigma^*$, a tokenizer $\mathcal{T} = (\tau, \kappa)$ from $\Sigma^*$ to $\Delta^*$ is* **consistent with respect to** *$p$ if we have $\kappa \tau p = p$.*

**Definition 3.3.** *Let $p$ be a probability distribution over $\Sigma^*$ and $\mathcal{T} = (\tau, \kappa)$ a tokenizer from $\Sigma^*$ to $\Delta^*$. When $\kappa \tau = \mathrm{id}_{\Sigma^*}$, we say that $\mathcal{T}$ is* **exact***.*

Notice that exact tokenizers are consistent, but a tokenizer that is consistent with respect to a distribution $p$ is not necessarily exact. Take, for instance, a probability distribution $p$ over some set $X$ and $x', x'' \in X$ such that $p(x') = p(x'') = c$. Then one can fashion a tokenizer for which $\kappa \tau(x) = x$ for all $x$ except $\kappa \tau(x') = x''$ and $\kappa \tau(x'') = x'$. Such a tokenizer is consistent with respect to $p$ without being exact. Consistency with respect to all distributions, however, is the same as being exact.

**Proposition 3.1.** *A tokenizer $\mathcal{T} = (\tau, \kappa)$ from $\Sigma^*$ to $\Delta^*$ is exact if and only if it is consistent with respect to every probability distribution over $\Sigma^*$.*

Our whole setting is quite general and can be represented by the following diagram:

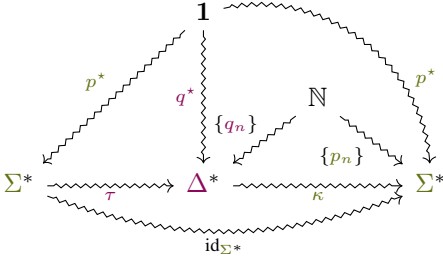

---

[3]The proofs for all formal results (theorems, propositions, lemmas) have been placed in the Appendix (§8).

When this diagram commutes, we have that $\kappa\tau p = p$ for all probability distributions $p$ over $\Sigma^*$, and Theorem 3.1 guarantees that $\{p_n\} \to p^\star$ when $\{q_n\} \to q^\star$.

Exact tokenizers have special properties. First, if a tokenizer $(\tau, \kappa)$ is exact, then $\kappa$ is deterministic over the image of $\tau$, because $\mathrm{id}_{\Sigma^*}$ also is. We make this notion formal in the following proposition.

**Proposition 3.2.** *Let $\mathcal{T} = (\tau, \kappa)$ be an exact tokenizer from $\Sigma^*$ to $\Delta^*$. Then $\kappa$ is deterministic on the support of $\tau$, i.e., $\kappa$ is deterministic $\tau$ almost everywhere.*

The condition $\kappa\tau = \mathrm{id}_{\Sigma^*}$ means $\tau$ is a right inverse (or **section**) of $\kappa$ and $\kappa$ is a left inverse (or **retraction**) of $\tau$. The proof of Proposition 3.2 shows that $\tau(\delta \mid \sigma)$ places probability zero on every $\delta \in \Delta^*$ such that $\kappa(\sigma \mid \delta) = 0$. Since $\kappa$ is deterministic in this case, it follows that, for an exact tokenizer $(\tau, \kappa)$, the encoder does not place positive probability mass on a token sequence for more than one text. In other words, $\tau$ is injective. Additionally, it must be that for each text $\sigma$ there is a $\delta$ with $\kappa(\sigma \mid \delta) = 1$, and therefore $\kappa$ is surjective.

## 4 STATISTICAL CONCERNS: INCONSISTENCY AND AMBIGUITY

While in most concrete cases of statistical language modeling a tokenizer's consistency is implicitly or explicitly assumed, there are many ways in which the conditions established in the previous section can, and in practice do, fail to be satisfied. In this section, we discuss two main statistical concerns to be considered when implementing or using tokenizers, namely inconsistency and ambiguity, and associate them with the properties of maps introduced in the previous section. The following definitions will be convenient.

**Definition 4.1.** *Given a tokenizer $\mathcal{T} = (\tau, \kappa)$, we say $\mathcal{T}$ has a **deterministic encoder** (resp. **decoder**) or is $\tau$-**deterministic** (resp. $\kappa$-**deterministic**) if $\tau$ (resp. $\kappa$) is a deterministic map. When a tokenizer $\mathcal{T}$ is both $\tau$-deterministic and exact, we have that $\mathcal{T}$ is also $\kappa$-deterministic, and $\kappa = \tau^{-1}$ over $\tau(\Sigma^*)$. Therefore, in such a case we say $\mathcal{T}$ is **bijective**.*

**Noninjective $\tau$ and Inconsistency.** Most commonly used tokenizers have deterministic encoders, including BPE (Sennrich et al., 2016) and WordPiece (Wu et al., 2016), as well as Unigram (Kudo, 2018) when used without regularization. As we have seen, functions can be understood as a particular case of stochastic maps where the probability mass is concentrated on one element. Tokenizers with deterministic encoders thus constitute a simplified form of tokenization. However, even in this simplified setting, the consistency of the tokenization process is not guaranteed. The following example offers an elementary intuition of this circumstance.

**Example 4.1.** *Consider the simple configuration represented in Fig. 1, where both $\tau$ and $\kappa$ are deterministic maps. Let $p^\star(\sigma_1) = 0.2$ and $p^\star(\sigma_2) = p^\star(\sigma_3) = 0.4$, with $p^\star(\sigma_i) = 0$ for $i > 3$. For $q^\star = \tau p^\star$, we have, therefore, $q^\star(\delta_1) = 0.2$, $q^\star(\delta_2) = 0$, and $q^\star(\delta_3) = 0.8$, with $q^\star(\delta_i) = 0$ for $i > 3$, and hence $\kappa\tau p^\star(\sigma_1) = 0 \neq 0.2$, $\kappa\tau p^\star(\sigma_2) = 0.2 \neq 0.4$, and $\kappa\tau p^\star(\sigma_1) = 0.8 \neq 0.4$. Assuming $\{q_n\}$ is a consistent estimator of $q^\star$, the pushforward of $q_n$ through $\kappa$ (i.e., $\kappa q_n$) would result in an inconsistent estimation of $p^\star$. Notice that the consistency of the tokenizer is relative to the distribution. Relative to a different distribution $p$ in $\Sigma^*$, where, for instance, $p(\sigma_1) = p(\sigma_2) = 0$ and $p = p^\star$ otherwise, the tokenizer specified in Fig. 1 is consistent. Hence the importance for a tokenizer to be exact.*

As shown in §3 and illustrated in Example 4.1, a fundamental cause of a tokenizer's inconsistency is the lack of injectivity of the encoder $\tau$. This is not just a theoretical concern. Even if in its abstract specification a tokenizer's encoder may appear to be injective, implementation decisions often introduce noninjective behaviors. These include normalizing operations, such as lowercasing, stripping accents, removing punctuation, or uniformizing whitespaces (e.g., Moi & Patry, 2023). Regardless of how the core tokenization function is defined, including this preprocessing step as part of the tokenizer model results in a noninjective encoding that compromises the consistency of estimators.

Even if text normalization is excluded from the decoding function, it can still happen that $\tau$ is undefined for some elements in $\Sigma$, and is, therefore, only a partial function. If the exceptions are handled by returning a unique distinguished token in $\Delta$ (e.g., an 'unknown' token unk), then $\tau$ becomes noninjective, incurring the risk of inconsistency. The appeal to an unk token and the difficulties associated with it have been widely studied from the perspective of OOV terms, especially

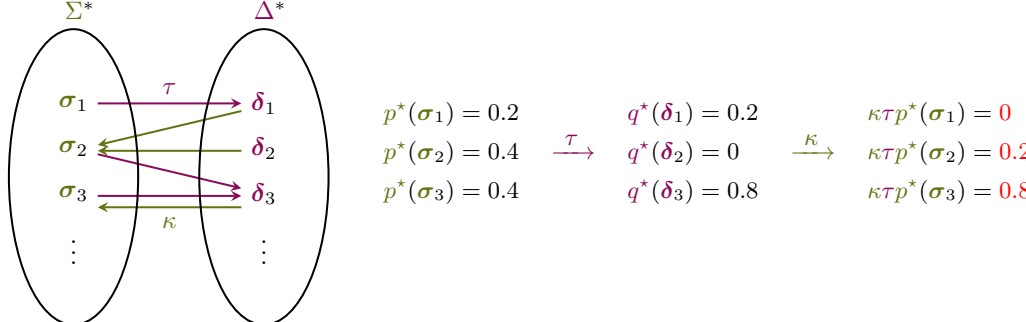

Figure 1: Example of an inconsistent tokenizer

in the context of NMT (e.g., Luong & Manning, 2016; Jean et al., 2015), ultimately leading to subword tokenizers as a way of providing "open vocabulary" solutions (Sennrich et al., 2016; Wu et al., 2016).[4] Formally, it is enough to inject $\Sigma$ into $\Delta$ (i.e., to include the alphabet in the vocabulary) to achieve an open vocabulary, something most tokenizers do by default. However, open vocabulary solutions do not entirely remove the risk of noninjective decoding. Some open vocabulary models, for instance, limit the size of $\Sigma$ to the sample's $k$ most frequent symbols, mapping all other symbols to an unk character *in* $\Sigma$ (e.g., Wu et al., 2016). Understood as a preprocessing step, this operation should not affect $\tau$'s injectivity. However, the use of copy models (e.g., Luong et al., 2015) that keep track of the original out-of-alphabet symbols to restore them in decoding, violates *de facto* the tokenizer's injectivity, and with it, the model's consistency over strings including those symbols.

Regardless of whether all symbols in the training sample are included in $\Sigma$, *out-of-alphabet* symbols can always be encountered at test or inference time. The recourse to a distinguished unk symbol *either in* $\Sigma$ *or* $\Delta$ must, therefore, be handled in such a way that the injectivity of $\tau$ is guaranteed. The use of a stochastic $\kappa$ (e.g., Mielke & Eisner, 2019) over the restricted domains of out-of-alphabet or out-of-vocabulary elements could, in principle, provide a novel way of addressing this problem in agreement with consistency concerns.

**Noninjective $\kappa$ and Ambiguity.** Whenever $\kappa$ is noninjective, the tokenizer introduces *ambiguity* in the model because more than one token sequence is mapped into a unique text. In bijective tokenizers, decoding is injective over the encoder's image, thus preventing ambiguity in principle. However, in practice, whenever $\tau(\Sigma^*)$ is a proper subset of $\Delta^*$, it may happen that the probability mass placed by the estimated language model outside the image of $\tau$ is nonzero, reintroducing ambiguity into the model (cf. Example 8.1 in the Appendix for an elementary illustration). This ambiguity is, however, **spurious** because $\tau$ was assumed to be deterministic, and hence the ambiguity does not originate from the reference distribution $p^\star$, but is a side effect of the estimator. An obvious source of spurious ambiguity lies in the fact that consistency is a property defined *in the limit*. As a consequence, for any $\delta \in \Delta^*$, $q_n(\delta)$ can and will generally differ from $q^\star(\delta)$. Spurious ambiguity can also result from the fact that, due to the properties of gradient descent and certain activation functions such as softmax, neural models are incapable of assigning zero probability to elements of $\Delta^*$. While spurious ambiguity has been identified among the motivations for introducing subword regularization (Kudo, 2018; Provilkov et al., 2020), it is often overlooked or disregarded, despite its potential nonnegligible effect on estimation (Cao & Rimell, 2021), although mostly in the case of "strongly out-of-domain evaluation sets" (Chirkova et al., 2023).

Spurious ambiguity is not the only kind of ambiguity that can result from the use of tokenization in language models. Whenever a tokenizer model is stochastic, a deterministic $\kappa$ must be noninjective for the model to preserve the consistency of estimators. However, the ambiguity thus introduced is not spurious in that it is deliberately designed for statistical purposes. In current tokenization practices, the main reason for the introduction of **stochastic** ambiguity is regularization (Kudo, 2018; Provilkov et al., 2020). The claim is that, by exhibiting different token sequences corresponding

---

[4]For a tokenization-free alternative to the OOV problem, see, for instance, Xue et al. (2022) and Clark et al. (2022), who also offer a good overview of existing approaches to this problem.

to the same text during training, a model increases its capability to handle text compositionality as well as its robustness to noise and tokenization errors. However, one could also conceive of a stochastic tokenizer where the possible images of a text reflect the objective probabilities of all **linguistic** ambiguities potentially affecting it (e.g., anⵏicecream, anⵏiceⵏcream, aⵏniceⵏcream as three possible token sequences for the text: a·n·i·c·e·c·r·e·a·m). This would require, however, to enhance tokenizers with linguistically motivated segmentation (see, for instance, Bostrom & Durrett, 2020; Hofmann et al., 2021; Gow-Smith et al., 2022; Beinborn & Pinter, 2023).

Although all these classes of ambiguity (spurious, stochastic, and linguistic) are both formally and semantically different, they all represent the same challenge for the tokenizer's consistency: The probability mass indirectly assigned by the model to one text in a language is spread over different token sequences. Notice that all these cases of ambiguity can coexist and, hence, their impact is difficult to evaluate. Yet, from a formal perspective, the solution for all these cases is the same: The computation of $\kappa q_n$ for a single text $\sigma \in \Sigma^*$ requires summing over all its preimages $\delta$ under $\kappa$, for which $q_n(\delta) > 0$, following the composition of stochastic maps presented in the previous section (Eq. (1)). However, such an operation can be computationally challenging because it can imply summing over a large or even infinite number of terms. For different strategies to address this challenge, see van Merriënboer et al. (2017); Buckman & Neubig (2018); Grave et al. (2019); Hannun et al. (2020); Cao & Rimell (2021); Vieira et al. (2024).

## 5 COMPUTATIONAL CONCERNS: FINITENESS AND SEQUENTIALITY

As the end of the previous section shows, even when a tokenizer model is consistent and all statistical concerns are taken into account, there are still computational aspects that can hinder the practice of tokenization. In this section, we turn to issues of finiteness and sequentiality.

**Multiplicativity and Finiteness.** Definitions 3.1 to 3.3 are general enough to allow for many kinds of encoding and decoding functions, including uncomputable ones; see Example 8.2 in the Appendix for an example. However, even when a tokenizer model is computable, its tractability is not guaranteed. Indeed, there are many reasons that could make the computation of tokenization intractable. Many of the operations that define tokenizer models involve sums over infinite sets. This is particularly true for the composition of stochastic maps whenever it is performed over an infinite domain, as in our case. Therefore, it is crucial to assess the tractability not only of $\tau$ and $\kappa$, but also of their composition $\kappa\tau$.

We have seen that when a tokenizer model is exact, $\tau$ is a section of $\kappa$, or equivalently, $\tau$ is injective. It follows that, for any $\sigma \in \Sigma^*$, $\tau$ concentrates the probability mass on only a subset of $\Delta^*$. This property can help reduce computational costs by restricting the sums to just those subsets. However, without further constraints, those subsets can still be infinite. For this reason, we introduce the following definitions.

**Definition 5.1.** *We say a tokenizer model $\mathcal{T} = (\tau, \kappa)$ is **multiplicative** if its decoder $\kappa$ respects the concatenation products, i.e., if $\kappa(\delta'\mathord{\shortmid}\delta'') = \kappa(\delta') \cdot \kappa(\delta'')$.*

**Definition 5.2.** *We say the **kernel** of a multiplicative tokenizer's decoder $\kappa$ is **trivial** if $\kappa$ maps nonempty token sequences to nonempty token sequences (i.e., if $\delta \neq \varepsilon_\Delta$ then $\kappa(\delta) \neq \varepsilon_\Sigma$).*

The most commonly used tokenizers, including BPE, WordPiece, and Unigram, are multiplicative. Notice that, for a kernel of a multiplicative tokenizer's decoder to be trivial, it is enough that $\kappa(\delta) \neq \varepsilon_\Sigma$ for any $\delta \in \Delta$. This implies that token sequences do not include special tokens that are erased during decoding. Importantly, when a multiplicative tokenizer's decoder has a trivial kernel, a decoded text cannot be shorter than the token sequence from which it has been decoded. This simple observation guarantees that the number of preimages of a text under $\kappa$ is finite. More precisely, we have the following proposition.

**Proposition 5.1.** *Let $\mathcal{T} = (\tau, \kappa)$ be a multiplicative tokenizer model whose decoder's kernel is trivial. For any $\delta \in \Delta^*$, $\kappa(\delta) = \sigma \implies |\delta| \leq |\sigma|$.*

**Corollary 5.0.1.** *Let $\mathcal{T} = (\tau, \kappa)$ be a multiplicative tokenizer model whose decoder's kernel is trivial. Then for any text $\sigma$, the set $\kappa^{-1}(\sigma)$ is finite.*

Proposition 5.1 guarantees that, if no token in the vocabulary is mapped to the empty string, then the length of every preimage of a text $\sigma$ of length $n$ has length less than or equal to $n$. So the

number of elements in $\kappa^{-1}(\sigma)$, and hence also the support of $\tau$ for any given $\sigma \in \Sigma^*$ when $(\tau, \kappa)$ is exact, is bounded by $\sum_{i=1}^{n} |\Delta|^i$. Since this bound is exponential in $n$, the exact or approximate computation of a tokenizer's encoding and decoding so that the consistency of the language model is not compromised requires the appeal to multiple strategies as the ones mentioned at the end of §4. Ideally, the complexity of a tokenizer model (that is, of the composition $\kappa\tau$) should be at most linear in the length of the input text. By placing all the probability mass on one token sequence, $\kappa\tau$ in exact deterministic tokenizers, such as BPE and WordPiece, can be computed in linear time as long as $\tau$ and $\kappa$ can be computed in linear time. However, rigorously handling spurious ambiguity still represents a challenge in these cases. In follow-up work, Vieira et al. (2024) exploit the finiteness of $\kappa^{-1}(\sigma)$ to introduce both exact and approximate algorithms for converting token-level language models to character-level ones, assuming $\kappa$ is **prefix monotone**, i.e., $\delta' \preceq \delta \implies \kappa(\delta') \preceq \kappa(\delta)$, a weaker condition implied by multiplicativity which is crucial in autoregressive models.[5]

**Bounded Variation and Sequentiality.** Finally, even though multiplicativity ensures that the length of every preimage of $\kappa^{-1}(\sigma)$ is bounded by the length of $\sigma$, the latter may still be unbounded. In practice, the bounded character of tokenization is secured externally by fixing a hyperparameter that artificially limits the length of input texts. However, boundedness can be addressed as an internal property of a tokenizer. For this purpose, we introduce the following definitions, adapted from Berstel (1979).

**Definition 5.3.** *The (left) distance between two strings $\gamma, \gamma' \in \Gamma^*$ is the number:*

$$\|\gamma, \gamma'\| \overset{\text{def}}{=} |\gamma| + |\gamma'| - 2\,|\gamma \wedge \gamma'|,$$

*where $\gamma \wedge \gamma'$ is the longest common prefix of $\gamma$ and $\gamma'$.*

**Definition 5.4.** *A function $f\colon \Gamma^* \to B^*$ has **bounded variation** if and only if $\forall k \geq 0,\ \exists C_k \geq 0$:*

$$\gamma, \gamma' \in \Gamma^*, \|\gamma, \gamma'\| \leq k \implies \|f(\gamma), f(\gamma')\| \leq C_k.$$

The importance of Definition 5.4 is that, following Choffrut's (1979) theorem, whenever $f$ preserves rational sets, if $f$ has bounded variation, then it is **subsequential**, that is, it can be realized by a deterministic finite-state transducer enhanced with a function over terminal states (cf. Definition 8.4 in the Appendix). Since subsequential functions are closed under composition (Berstel, 1979, prop. IV.2.5), for $\kappa\tau$ to be subsequential, it is enough that both $\tau$ and $\kappa$ are. Given that most commonly used tokenizers are multiplicative, the following result is significant.

**Proposition 5.2.** *If a function $f\colon \Gamma^* \to B^*$ is multiplicative then it has bounded variation.*

However, in the context of tokenizers, multiplicativity concerns only the decoder $\kappa$ (a multiplicative encoder $\tau$ is not a desirable feature for a tokenizer), thus shifting the focus towards the encoder $\tau$. The "maximal munch" approach (Reps, 1998; Palmer, 2000) adopted by WordPiece, for instance, iteratively maps the successive longest prefixes of a text to tokens in the vocabulary. WordPiece's encoder is thus bounded by the maximum length of the preimages of $\Delta$ under $\tau$, and is, therefore, subsequential (cf. Song et al., 2021, for a realization of WordPiece by a finite-state transducer). Moreover, assuming specific conditions on the structure of the list of rules or "merges", Berglund & van der Merwe (2023) and Berglund et al. (2024) proposed an algorithm for constructing deterministic finite automata realizing BPE's encoder, thus suggesting that, under such conditions, BPE is also subsequential.

## 6 CONCLUSION

In this work, we have addressed the use of token representations in NLP from a foundational perspective. Relying on the category of stochastic maps as an elementary formal tool, we proposed a general definition of a tokenizer as an arbitrary pair of composable maps. The framework proposed enabled us to formally establish several properties of tokenization, and most importantly, the necessary and sufficient condition for a tokenizer to preserve the consistency of estimators. Furthermore, our approach allowed to shed new theoretical light on known issues concerning tokenization, namely inconsistency, ambiguity, finiteness, and sequentiality, by characterizing the latter through formal properties of composable maps such as injectivity, multiplicativity, or bounded variation. We believe this framework will inform future empirical research and contribute to establishing and developing theoretical and practical aspects of representation learning in NLP on solid grounds, especially in cases where the reliance on a model requires to go beyond its mere performance, and take into account properties such as formal guarantees, verification, theoretical soundness, interpretability, or liability.

---

[5]Note that Vieira et al. (2024) call "non-erasing" the fact that $\kappa$ has a trivial kernel.

## 7 ACKNOWLEDGEMENTS

The authors thank Saeed Zakeri for pointing out an interesting interplay between a general fact about the Banach spaces $L^1(X)$ for any measure space $X$ and the fact that for the little $l^p$ spaces, the $l^\infty$ norm is smaller than the $l^1$ norm, which led to the proof of Lemma 3.1. The authors also thank Li Du for useful feedback at the early stages of this work.

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

## 8 APPENDIX

### 8.1 PROOFS

**Lemma 3.1.** *Let $\{p_n\}$ be a sequence of probability distributions over a countable set $X$ that converges pointwise to a probability distribution $p$. Then $\lim_{n\to\infty} \sum_{x\in X} |p_n(x) - p(x)| = 0$.*

*Proof.* Fatou's lemma applied to $X$ with the counting measure implies that for any sequence of nonnegative functions $\{f_n\}$ on $X$,

$$\sum_{x\in X} \liminf_{n\to\infty} f_n(x) \le \liminf_{n\to\infty} \sum_{x\in X} f_n(x). \tag{2}$$

We will apply this to $f_n := p_n + p - |p_n - p|$. First, note that since $\lim_{n\to\infty} p_n(x) = p(x)$, we have $\liminf_{n\to\infty} f_n(x) = p(x) + p(x) - 0 = 2p(x)$ so the left hand side of equation 2 becomes $\sum_{x\in X} 2p(x) = 2$. Therefore,

$$2 \le \liminf_{n\to\infty} \sum_{x\in X} f_n(x) \tag{3a}$$

$$= \liminf_{n\to\infty} \sum_{x\in X} p_n(x) + p(x) - |p_n(x) - p(x)| \tag{3b}$$

$$= \liminf_{n\to\infty} \sum_{x\in X} p_n(x) + \sum_{x\in X} p(x) - \sum_{x\in X} |p_n(x) - p(x)| \tag{3c}$$

$$= \liminf_{n\to\infty} 1 + 1 - \sum_{x\in X} |p_n(x) - p(x)| \tag{3d}$$

$$= 2 - \limsup_{n\to\infty} \sum_{x\in X} |p_n(x) - p(x)|. \tag{3e}$$

It follows that $\limsup_{n\to\infty} \sum_{x\in X} |p_n(x) - p(x)| \le 0$, so $\lim_{n\to\infty} \sum_{x\in X} |p_n(x) - p(x)| = 0$. $\square$

**Corollary 3.0.1.** *Let $\{p_n\}$ be a sequence of probability distributions over a countable set $X$ that converges pointwise to a probability distribution $p$. Then $\{p_n\} \to p$ uniformly.*

*Proof.* Since the sum of nonnegative numbers is always greater than any particular term in the sum and $\lim_{n\to\infty} \sum_{x\in X} |p_n(x) - p(x)| = 0$, we can conclude that the sequence $\{p_n\} \to p$ uniformly. $\square$

**Lemma 3.2.** *Let $f$ be a stochastic map from $X$ to $Y$, and $\{p_n\}$ be an estimator for a probability distribution $p$ on $X$. Then $fp_n$ is an estimator for the probability distribution $fp$ on $Y$.*

*Proof.* Fix $y \in Y$. We will show that $\{fp_n(y)\} \to fp(y)$. By Lemma 3.1, we have that $\lim_{n\to\infty} \sum_{x\in X} |p_n(x) - p(x)| = 0$. Therefore,

$$\lim_{n\to\infty} |fp_n(y) - fp(y)| = \lim_{n\to\infty} \left| \sum_{x\in X} f(y \mid x)p_n(x) - \sum_{x\in X} f(y \mid x)p(x) \right| \tag{4a}$$

$$\le \lim_{n\to\infty} \sum_{x\in X} f(y \mid x) |p_n(x) - p(x)| \tag{4b}$$

$$\le \lim_{n\to\infty} \sum_{x\in X} |p_n(x) - p(x)| \tag{4c}$$

$$= 0. \tag{4d}$$

$\square$

**Theorem 3.1** (Fundamental Principle of Tokenization)**.** *Given a reference probability distribution $p^\star$ over $\Sigma^*$, a tokenizer $\mathcal{T} = (\tau, \kappa)$ from $\Sigma^*$ to $\Delta^*$, and a consistent estimator $\{q_n\}$ of the image reference distribution $q^\star = \tau p^\star$, the sequence $\{\kappa q_n\}$ is a consistent estimator of $p^\star$ if and only if $\kappa\tau p^\star = p^\star$.*

*Proof.* By hypothesis, $\{q_n\} \to q^\star$ and by definition $q^\star = \tau p^\star$. By Lemma 3.2, applying $\kappa$ to both sides, we have that $\{\kappa q_n\} \to \kappa q^\star$ and so

$$\{\kappa q_n\} \to \kappa \tau p^\star.$$

Therefore, if $\kappa \tau p^\star = p^\star$ we have $\{\kappa q_n\} \to p^\star$. Conversely, if $\{\kappa q_n\} \to p^\star$ we have both $\{\kappa q_n\} \to p^\star$ and $\{\kappa q_n\} \to \kappa \tau p^\star$ and so by the uniqueness of limits, $\kappa \tau p^\star = p^\star$. $\qquad\square$

**Proposition 3.1.** *A tokenizer $\mathcal{T} = (\tau, \kappa)$ from $\Sigma^*$ to $\Delta^*$ is exact if and only if it is consistent with respect to every probability distribution over $\Sigma^*$.*

*Proof.* Exact means $\kappa\tau = \mathrm{id}_{\Sigma^*}$ hence $\kappa\tau p = p$ for every probability distribution $p$ on $\Sigma^*$. To prove the other direction, suppose that $\kappa\tau p = p$ for every $p$ on $\Sigma^*$. Fix an arbitrary $\sigma \in \Sigma^*$. Let $p_\sigma$ be the point mass distribution on $\Sigma^*$ concentrated on $\sigma$. So $p_\sigma(\sigma) = 1$ and $p_\sigma(\sigma') = 0$ for any $\sigma' \neq \sigma$. By hypothesis, $p_\sigma = \kappa\tau p_\sigma$. Apply to $\sigma$ to get $1 = \kappa\tau p_\sigma(\sigma)$. The right hand side, as the pushforward of $p_\sigma$ via $\kappa\tau$, says $1 = p_\sigma(\kappa\tau(\sigma))$. Since $p_\sigma$ takes the value 1 at only one point, it follows that the argument $\kappa\tau(\sigma) = \sigma$. Since $\sigma$ was arbitrary, we conclude that $\kappa\tau = \mathrm{id}_{\Sigma^*}$, i.e., the tokenizer $(\tau, \kappa)$ is exact. $\qquad\square$

**Proposition 3.2.** *Let $\mathcal{T} = (\tau, \kappa)$ be an exact tokenizer from $\Sigma^*$ to $\Delta^*$. Then $\kappa$ is deterministic on the support of $\tau$, i.e., $\kappa$ is deterministic $\tau$ almost everywhere.*

*Proof.* Assume $(\tau, \kappa)$ is exact, i.e., $\kappa\tau = \mathrm{id}_{\Sigma^*}$, and let $\sigma \in \Sigma^*$.

Since $\sum_{\delta \in \Delta^*} \tau(\delta \mid \sigma) = 1$ and $\kappa\tau(\sigma|\sigma) = \mathrm{id}_{\Sigma^*}(\sigma \mid \sigma) = 1$. We obtain:

$$0 = \sum_{\delta \in \Delta^*} \tau(\delta \mid \sigma) - \kappa\tau(\sigma|\sigma) \tag{5a}$$

$$= \sum_{\delta \in \Delta^*} \tau(\delta \mid \sigma) - \sum_{\delta \in \Delta^*} \kappa(\sigma \mid \delta)\tau(\delta \mid \sigma) \tag{5b}$$

$$= \sum_{\delta \in \Delta^*} \tau(\delta \mid \sigma) - \kappa(\sigma \mid \delta)\tau(\delta \mid \sigma) \tag{5c}$$

$$= \sum_{\delta \in \Delta^*} \tau(\delta \mid \sigma)(1 - \kappa(\sigma \mid \delta)). \tag{5d}$$

Since Eq. (5d) is a sum of nonnegative terms that equals zero, each terms must be zero. It follows that if $\tau(\delta \mid \sigma) > 0$ for some $\delta$ (e.g.: $\delta$ is in the support of $\tau$) then then $1 - \kappa(\sigma \mid \delta) = 0 \Leftrightarrow \kappa(\sigma \mid \delta) = 1$. From which it follows that

$$\kappa(\sigma' \mid \delta) = \begin{cases} 1 & \text{if } \sigma' = \sigma \\ 0 & \text{if } \sigma' \neq \sigma. \end{cases}.$$

$\qquad\square$

**Proposition 5.1.** *Let $\mathcal{T} = (\tau, \kappa)$ be a multiplicative tokenizer model whose decoder's kernel is trivial. For any $\delta \in \Delta^*$, $\kappa(\delta) = \sigma \implies |\delta| \leq |\sigma|$.*

*Proof.* We can reason by induction. Let $|\delta| = m$ and $|\sigma| = n$. The property is true when $m = 1$ since 1 is the minimun length of any possible image of $\kappa$. Assume it is true for $m = k$ and let $\delta = \delta_1 | \ldots | \delta_k | \delta_{k+1}$. Then $\kappa(\delta) = \kappa(\delta_1 | \ldots | \delta_k) \cdot \kappa(\delta_{k+1}) = \sigma_1 \cdot \ldots \cdot \sigma_r \cdot \sigma_{r+1} \cdot \ldots \cdot \sigma_{r+s}$, where $\kappa(\delta_{k+1}) = \sigma_{r+1} \cdot \ldots \cdot \sigma_{r+s}$. Since $r \geq k$ and $s \geq 1$, we have that $r + s \geq k + 1$. $\qquad\square$

**Proposition 5.2.** *If a function $f : \Gamma^* \to B^*$ is multiplicative then it has bounded variation.*

*Proof.* Assume $f$ is multiplicative and let $k$ be given. Suppose $\gamma, \gamma' \in \Gamma^*$ satisfy $\|\gamma, \gamma'\| \leq k$. Let

$$C_k = \max_{\substack{\gamma'', \gamma''' \in \Gamma^* \\ |\gamma''| + |\gamma'''| \leq k}} \{|f(\gamma'')| + |f(\gamma''')|\}.$$

Write $\gamma$ and $\gamma'$ as $\gamma = \gamma \wedge \gamma' \cdot \gamma''$ and $\gamma' = \gamma \wedge \gamma' \cdot \gamma'''$. Notice that $\|\gamma'', \gamma'''\| = |\gamma''| + |\gamma'''| \leq k$, and look at $\|f(\gamma), f(\gamma')\|$:

$$\|f(\gamma), f(\gamma')\| = \|f(\gamma \wedge \gamma' \cdot \gamma''), f(\gamma \wedge \gamma' \cdot \gamma''')\| \tag{6a}$$

$$= \|f(\gamma \wedge \gamma') \cdot f(\gamma''), f(\gamma \wedge \gamma') \cdot f(\gamma''')\| \tag{6b}$$

$$= \|f(\gamma''), f(\gamma''')\| \tag{6c}$$

$$\leq |f(\gamma'')| + |f(\gamma''')| \tag{6d}$$

$$\leq C_k \tag{6e}$$

$\square$

## 8.2 EXAMPLES

**Example 8.1.** *Take, for instance, a bijective tokenizer such as BPE or WordPiece, with $\kappa$ performing concatenation of the token maps in the usual way. Let $\Sigma = \{\mathsf{t}, \mathsf{h}, \mathsf{e}\}$ and $\Delta = \{\mathsf{t}, \mathsf{h}, \mathsf{e}, \mathsf{th}, \mathsf{he}\}$. In this minimal configuration, it is easy to see that $\kappa(\mathsf{t|h|e}) = \kappa(\mathsf{t|he}) = \kappa(\mathsf{th|e}) = \mathsf{t} \cdot \mathsf{h} \cdot \mathsf{e} \in \Sigma^*$. However, BPE or WordPiece being bijective tokenizers, $\tau$ can only map the value of $\kappa$ to at most one of the latter's arguments, say $\tau(\mathsf{t} \cdot \mathsf{h} \cdot \mathsf{e}) = \mathsf{th|e}$. We then have that $\tau(\kappa(\mathsf{t|he})) \neq \mathsf{t|he}$ (and likewise for $\mathsf{t|h|e}$). If the estimator happens to place nonzero probability mass on any of the latter two token sequences, the model will exhibit spurious ambiguity.*

**Example 8.2.** *Let $\Sigma = \Delta = \{\mathsf{0}, \mathsf{1}\}$, and define $\mathcal{T}_{unc} = (\tau_{unc}, \kappa_{unc})$ as a deterministic model in the following way:*

$$\tau_{unc}(\boldsymbol{\sigma}) = \begin{cases} \boldsymbol{\sigma}|\mathsf{1}, & \text{if } \boldsymbol{\sigma} \text{ describes a valid} \\ & \text{Turing Machine followed by} \\ & \text{an input for which it halts.} \\ \boldsymbol{\sigma}|\mathsf{0}, & \text{otherwise.} \end{cases} \qquad \kappa_{unc}(\boldsymbol{\delta}) = \begin{cases} \varepsilon, & \text{if } \boldsymbol{\delta} \in \Delta. \\ \boldsymbol{\sigma}, & \text{otherwise, where } \boldsymbol{\delta} = \boldsymbol{\sigma}|\delta. \end{cases}$$

*Significantly, $\mathcal{T}_{unc}$ is not only well-defined but also exact and therefore consistent for any language model $p$ over $\Sigma^*$. However, $\tau_{unc}$ is famously an uncomputable function, and hence $\mathcal{T}_{unc}$ is an uncomputable tokenizer.*

## 8.3 DEFINITIONS

**Definition 8.1.** *A **(left) sequential transducer** is a 6-tuple $(Q, \Gamma, \mathrm{B}, i, \diamond, *)$, where:*

| | |
|---|---|
| $Q$ | *is a set of states* |
| $\Gamma$ | *is an input alphabet* |
| $\mathrm{B}$ | *is an output alphabet* |
| $i \in Q$ | *is an initial state* |
| $\diamond \colon D \subseteq Q \times \Gamma \to Q$ | *is an input or "next state" function* |
| $* \colon D \subseteq Q \times \Gamma \to \mathrm{B}^*$ | *is an output function.* |

**Definition 8.2.** *A function $f \colon \Gamma^* \to \mathrm{B}^*$ is **sequential** if it is realized by a sequential transducer, i.e., $f(\boldsymbol{\gamma}) = (i * \boldsymbol{\gamma})$.*

**Definition 8.3.** *A **(left) subsequential transducer** is a 7-tuple $(Q, \Gamma, \mathrm{B}, i, \diamond, *, \rho)$, where:*

| | |
|---|---|
| $(Q, \Gamma, \mathrm{B}, i, \diamond, *)$ | *is a sequential transducer* |
| $\rho \colon Q \to \mathrm{B}^*$ | *is a terminal function.* |

**Definition 8.4.** *A function $f \colon \Gamma^* \to \mathrm{B}^*$ is **subsequential** if it is realized by a subsequential transducer $(Q, \Gamma, \mathrm{B}, i, \diamond, *, \rho)$, i.e., $f(\boldsymbol{\gamma}) = (i * \boldsymbol{\gamma}) \cdot \rho(i \diamond \boldsymbol{\gamma})$.*

