# OpenReview forum: "The Foundations of Tokenization: Statistical and Computational Concerns"
_ICLR.cc/2025/Conference — ICLR 2025 Poster_

### Official Review · Reviewer_btSn · 2024-10-24

**Soundness:** 3
**Presentation:** 2
**Contribution:** 2
**Rating:** 5
**Confidence:** 3

**Summary:**

The paper investigates into the tokenization process for language modelling from a theoretical viewpoint.

The framework examines language modelling as mapping between a true language model on an alphabet and a learned language model on a separate alphabet, known as a stochastic map. In section 3, the paper establishes convergence property for various distributions in different spaces, as well as necessary function properties (injectivity and surjectivity). Later on, the paper leverages such properties to examine the problems (e.g injectivity, surjectivity) of current language models and connect them with concrete issues such as inconsistency, ambiguity, tractability, etc.

**Strengths:**

The paper proposes an interesting and also promising theoretical direction to examine the tokenization problems: by explicitly examining the relationship between the true language model under its alphabet and the learned language model with the practical alphabet. Under this framework, the paper has shown some interesting preliimary results:
- Define propery the consistency from analysis viewpoint
- The propoerty of encoder and decoder and connect them with the problem of inconsistency and ambiguity

**Weaknesses:**

The connections between the derived properties/theorems and the NLP practice/problems are not strong. My arguments are stated as the following:
- In theorem 3.1 the authors derive the condition to make the estimator consistent; however, this property is nowhere more leveraged, the paper discusses in more detail the exact mappings and the derived injectivity/surjectivity properties which do not require the analysis from 3.1 and makes 3.1 a isolated construction without any implication built inside the paper.
- The injectivity (and surjectivity) discussions look more like discussions as a "related work" section. For example, no derivations are shown for problems such as "how severe the problem is when injectivity assumption is broken" or "when injectivity are ensured by certain techniques, what is the advantage quantified/bounded mathematically".
- While the boundness and tractability discussions are interesting, they are of observational nature. The paper will have significantly more impact if the paper is able to derive algorithms/improvements/theoretical implication out of them.

**Questions:**

Can the authors further clairify the connection between 3.1 and the rest of the paper please?

---

> ### Author Response · Authors · 2024-11-23
>
> Thank you very much for your careful evaluation and remarks. We appreciate your comments on the paper’s strengths.
>
> We are also grateful for pointing out that the connection between Theorem 3.1 and the remaining sections of the paper is not clear. The connection is, however, essential in our view. As stated at the beginning of section 4, while consistency is usually assumed, the necessary conditions for consistency formally established in th 3.1 are not usually met. The entire section is therefore a way to show that known problems commonly addressed from a practical or empirical perspective (e.g., OOV terms, special tokens, ambiguity, approximation errors, positivity of the softmax activation function, etc.) are in fact related to formal properties that violate the conditions formally established in th. 3.1. These conditions are precisely those related to injectivity and surjectivity of $\tau$ and $\kappa$. If section 4 discusses injectivity and surjectivity, it is because these are formal consequences of exactness, which is, in turn, an unconditioned form to obtain consistency (Prop. 3.1). So, if the discussed conditions on injectivity and surjectivity are not met, consistency is not guaranteed. As for section 5, the idea is that while consistency is not everything that matters, the use of the formal properties of $\tau$ and $\kappa$ laid down to establish its necessary and sufficient conditions can be further used to analyze and guarantee computational requirements. We hope this clarification can help to see the connection more clearly. If you have any suggestions on how this can become more clear in the paper, we would be happy to include them in a final version.
>
> We agree that this paper does not provide any empirical results concerning the connection between these formal properties and their practical consequences. We also agree that those results would significantly contribute to enhancing the paper’s impact, even if we believe that the purely theoretical character of the paper shouldn’t be a weakness *per se* in the ICLR area of “Learning theory” where it has been submitted. On this point, we would like to provide some context. This paper constitutes the first step in a broader research program aiming at addressing tokenization problems from a formal principled perspective, including various empirical results on practical issues. Given the strict page limit of ML and NLP conferences, it was impossible for us to include all the aspects and results of this program in one paper. Therefore, we split up the work over multiple papers, distributing their content as consistently as we could. Concretely, we have used the formalism introduced in this paper to design algorithms for converting token-level autoregressive language models to character-level ones, present both exact and approximate algorithms, and benchmark their practical runtime and approximation quality. The framework proposed here (especially, the principles discussed in the last section) was crucial to identify a minimal subset of $\Delta^*$ over which an exact, tractable algorithm could be defined, as well as a consistent notion of approximation, all of which uses injective, surjective, multiplicative, and kernel properties of the maps $\tau,\kappa$. Direct practical applications of this work include: principled solutions to the prompt boundary problem, constrained generation expressed over characters, or consistent linguistic and psycholinguistic measures. Concerning this last point, for instance, we have already performed an analysis of the role of tokenizers in psycholinguistics, providing an exact way of computing the surprisal of regions of interest over characters from a token model to assess psycholinguistic measures of reading times (this work has been recently published). Currently, we are studying the preservation (or lack thereof) by tokenization of structural features induced by formal grammars over character and token models (work in progress). All these applications use the framework proposed here as their theoretical foundation. Accordingly, we thought that the most consistent way to disseminate the different steps in our program was to adopt a “division of labor” between papers, keeping this paper in its pure theoretical form without tying it to any one application, and envisaging its impact as part of the broader research program.

---

> > ### Comment · Reviewer_btSn · 2024-11-23
> >
> > I appreciate the authors' feedback which gives further clarifications on the concerns I have. While I agree that the theoretic research that this paper follows is interesting, the general theorem (3.1) does not relate to tokenization in any particular way and is a general math derivations. The paper would benefit from further content/materials/empirical analysis and I personally think it is a pity that the authors choose to split such things out of the content of this paper, rending the current paper below acceptance criteria.

---

> > > ### Author Response · Authors · 2024-11-23
> > > **Clarification**
> > >
> > > Thank for your feedback. Could we ask for a quick clarification on how Theorem 3.1 does not relate to tokenization in any particular way? It tells us when we can consistently estimate a language model under a tokenizer.

---

### Official Review · Reviewer_rVcs · 2024-11-01

**Soundness:** 3
**Presentation:** 3
**Contribution:** 2
**Rating:** 5
**Confidence:** 1

**Summary:**

The authors address the problems of tokenization and use of the token representations in NLP from a foundational mathematical perspective. Based on the category of stochastic maps, the authors propose a general definition of a tokenizer as an arbitrary pair of composable maps, and this theoretical framework enables to establish general conditions for a principled use of tokenizers and the necessary and sufficient conditions for a tokenizer model to preserve the consistency of statistical estimators. The statistical and computational concerns (inconsistency, ambiguity, tractability, and boundedness) are discussed. To achieve this, authors characterize these known issues through the lens of formal properties of composable maps, such as injectivity, multiplicativity, and finite decomposition.

**Strengths:**

— Theoretical position paper is from a foundational perspective, and mathematically justified analysis is introduced; strict definitions and notation conventions are presented.

**Weaknesses:**

It's still unclear how to apply this knowledge to practical issues and better tokenizers for LLMs.

**Questions:**

I am unsure if I am not a relevant reviewer for this paper or if the paper is too foundational for the conference.

---

> ### Author Response · Authors · 2024-11-23
>
> Thank you very much for your evaluation and remarks. We appreciate your comments on the paper’s strengths.
>
> We would like to provide some context concerning the foundational character of this paper and its relation to practical issues. This is indeed a completely theoretical paper, submitted in the principal ICLR area of “Learning theory”, for which we believe the paper is a good match. We think that the purely theoretical character of a paper shouldn’t be a weakness *per se* in this area. We agree that associating the proposed theoretical framework with practical aspects of tokenization would significantly contribute to enhancing its impact. In this regard, we would like to mention that this paper constitutes the first step in a broader research program aiming at addressing tokenization problems from a formal, principled perspective, including various practical issues. Given the strict page limit of ML and NLP conferences, it was impossible for us to include all the aspects and results of this program in one paper. Therefore, we split up the work over multiple papers, distributing their content as consistently as we could. Concretely, we have used the formalism introduced in this paper to design algorithms for converting token-level autoregressive language models to character-level ones, present both exact and approximate algorithms, and benchmark their practical runtime and approximation quality. The framework proposed here (especially, the principles discussed in the last section) was crucial to identify a minimal subset of $\Delta^*$ over which an exact, tractable algorithm could be defined, as well as a consistent notion of approximation, all of which uses injective, surjective, multiplicative, and kernel properties of the maps $\tau,\kappa$. Direct practical applications of this work include: principled solutions to the prompt boundary problem, constrained generation expressed over characters, or consistent linguistic and psycholinguistic measures. Concerning this last point, for instance, we have already performed an analysis of the role of tokenizers in psycholinguistics, providing an exact way of computing the surprisal of regions of interest over characters from a token model to assess psycholinguistic measures of reading times (this work has been recently published). Currently, we are studying the preservation (or lack thereof) by tokenization of structural features induced by formal grammars over character and token models (work in progress). All these applications use the framework proposed here as their theoretical foundation. Accordingly, we thought that the most consistent way to disseminate the different steps in our program was to adopt a “division of labor” between papers, keeping this paper in its pure theoretical form without tying it to any one application, and envisaging its impact as part of the broader research program.

---

### Official Review · Reviewer_pg9p · 2024-11-01

**Soundness:** 4
**Presentation:** 4
**Contribution:** 3
**Rating:** 8
**Confidence:** 3

**Summary:**

This paper presents a formalization of the tokenization process in modern language models. The formalization is based on the category of stochastic maps, which presents a novel and unified framework for representing and analyzing tokenizers. The paper also provides some examples of sing the framework to account for tokenizer behavior such as ambiguity and inconsistencies.

**Strengths:**

The paper presents a novel framework for representing and analyzing tokenizers in the form of stochastic maps. The proposed framework has the potential to provide more a foundational statistical understanding of tokenizer behavior, which may lead to practical improvements.

**Weaknesses:**

The paper provides some examples of how the framework can be utilized to shed new light on tokenizer behavior such as ambiguity and inconsistency, but it is not immediately clear to what extent the proposed framework can lead to practical improvements of tokenizer performance.

**Questions:**

Would it be possible to provide some concrete examples of how the framework can be used to shed new light on differences between different types of tokenizers?

---

> ### Author Response · Authors · 2024-11-23
>
> Thank you very much for your careful evaluation and remarks. We appreciate your comments on the paper’s strengths.
>
> Thanks also for your remark and question concerning the practical consequences of the framework proposed. This is indeed a completely theoretical paper, submitted in the principal area of “Learning theory”. We agree that associating the proposed theoretical framework with practical aspects of tokenization would significantly contribute to enhancing its impact. On this point, we would like to provide some context that might help explain our choices. This paper constitutes the first step in a broader research program aiming at addressing tokenization problems from a formal principled perspective, including various practical issues. Given the strict page limit of ML and NLP conferences, it was impossible for us to include all the aspects and results of this program in one paper. Therefore, we split up the work over multiple papers, distributing their content as consistently as we could. While improving the performance is indeed one possible practical outcome of our framework, we believe there are other practical outcomes. Concretely, we have used the formalism introduced in this paper to design algorithms for converting token-level autoregressive language models to character-level ones, present both exact and approximate algorithms, and benchmark their practical runtime and approximation quality. The framework proposed here (especially, the principles discussed in the last section) was crucial to identify a minimal subset of $\Delta^*$ over which an exact, tractable algorithm could be defined, as well as a consistent notion of approximation, all of which uses injective, surjective, multiplicative, and kernel properties of the maps $\tau,\kappa$. Direct practical applications of this work include, among others, principled solutions to the prompt boundary problem, constrained generation expressed over characters, or consistent linguistic and psycholinguistic measures. Concerning this last point, for instance, we have already performed an analysis of the role of tokenizers in psycholinguistics, providing an exact way of computing the surprisal of regions of interest over characters from a token model to assess psycholinguistic measures of reading times (this work has been recently published). Currently, we are studying the preservation (or lack thereof) by tokenization of structural features induced by formal grammars over character and token models (work in progress). All these applications use the framework proposed here as their theoretical foundation. Accordingly, we thought that the most consistent way to disseminate the different steps in our program was to adopt a “division of labor” between papers, keeping this paper in its pure theoretical form without tying it to any one application, and envisaging its impact as part of the broader research program.

---

### Official Review · Reviewer_yPjZ · 2024-11-07

**Soundness:** 3
**Presentation:** 4
**Contribution:** 3
**Rating:** 8
**Confidence:** 3

**Summary:**

Disclaimer: I previously reviewed an earlier version of this paper during its submission to NeurIPS 2024.

This paper proposes a formal approach to tokenization. Specifically, the authors apply the notion of stochastic maps (Baez & Fritz, 2014) to tokenization models, which allows them to formalize key properties of tokenizers such as their consistency. In a next step, they use their framework to analyze further statistical and computational aspects of tokenization (e.g., spurious ambiguity).

**Strengths:**

The paper has several strengths:

- Tokenization is a critical aspect of modern-day natural language processing, but its theoretical underpinnings are not yet fully understood. The formalisms introduced in the paper help close this gap and might become the basis for future work.
- The application of stochastic maps to tokenization is novel.
- The presentation is excellent; the writing is clear and overall easy to follow.

**Weaknesses:**

This is a completely theoretical paper without any empirical evaluation. While not a weakness _per se_, the authors mention that their findings have implications for the practical use of tokenizers (e.g., line 95). Compared to the version of the paper that was under submission at NeurIPS 2024, the authors have added sections discussing practical aspects of their observations (e.g., lines 345-350), but I still believe that a proper case study showcasing the practical value of the proposed formal framework would greatly enhance the potential impact of the paper. This is particularly desirable since many of the discussed problems are well-known in the community, so it is not clear what exactly the paper adds beyond a new theoretical perspective.

**Questions:**

Comments:
- While you mention the relation between tokenization and linguistic segmentation, you should give more credit to the line of work that has investigated this link over the last few years (e.g., [Bostrom & Durrett, 2020](https://aclanthology.org/2020.findings-emnlp.414/), [Hofmann et al., 2021](https://aclanthology.org/2021.acl-long.279/), [Gow-Smith et al., 2022](https://aclanthology.org/2022.emnlp-main.786/), [Beinborn & Pinter, 2023](https://aclanthology.org/2023.emnlp-main.272/)), which is also relevant for your discussion of linguistic ambiguities (lines 395-399).
- Lines 405-410: I think you should mention that the practical benefit of this operation has been shown to be negligible in most cases ([Chirkova et al., 2023](https://aclanthology.org/2023.acl-short.1/)).
- I would recommend putting the examples from the appendix back into the main paper, especially Example 8.1. You have a bit of space left, so this should be feasible, and it would help illustrate your arguments.

Typos:
- Line 120: "distributions" -> "distribution"
- Line 158: "equation equation 1" -> "equation 1"
- Line 205: "this" -> "these"
- Line 459: "complexity a tokenizer model" -> "complexity of a tokenizer model"

---

> ### Author Response · Authors · 2024-11-23
>
> Thank you very much for your careful evaluation and remarks. We appreciate your comments on the paper’s strengths.
>
> This is indeed a completely theoretical paper, and we share the view that this should not constitute *a priori* a weakness, especially for a submission in the principal area of “Learning theory” as this one. We are also grateful for mentioning the improvements in this sense compared to our previous version. That being said, we agree that associating the proposed theoretical framework with practical aspects of tokenization would significantly contribute to enhancing its impact. On this point, we would like to provide some context that might help explain our choices. This paper constitutes the first step in a broader research program aiming at addressing tokenization problems from a formal principled perspective. Given the strict page limit of ML and NLP conferences, it was impossible for us to include all the aspects and results of this program in one paper. Therefore, we split up the work over multiple papers, distributing their content as consistently as we could. Concretely, we have used the formalism introduced in this paper to design algorithms for converting token-level autoregressive language models to character-level ones, present both exact and approximate algorithms, and benchmark their practical runtime and approximation quality. The framework proposed here (especially, the principles discussed in the last section) was crucial to identify a minimal subset of $\Delta^*$ over which an exact, tractable algorithm could be defined, as well as a consistent notion of approximation, all of which uses injective, surjective, multiplicative, and kernel properties of the maps $\tau,\kappa$. Direct practical applications of this work include: principled solutions to the prompt boundary problem, constrained generation expressed over characters, or consistent linguistic and psycholinguistic measures. Concerning this last point, for instance, we have already performed an analysis of the role of tokenizers in psycholinguistics, providing an exact way of computing the surprisal of regions of interest over characters from a token model to assess psycholinguistic measures of reading times (this work has been recently published). Currently, we are studying the preservation (or lack thereof) by tokenization of structural features induced by formal grammars over character and token models (work in progress). All these applications use the framework proposed here as their theoretical foundation. Accordingly, we thought that the most consistent way to disseminate the different steps in our program was to adopt a “division of labor” between papers, keeping this paper in its pure theoretical form without tying it to any one application, and envisaging its impact as part of the broader research program.
>
> We agree that more credit should be attributed to the work mentioned in the points suggested. We added remarks concerning the relation of tokenizers to linguistic segmentation with reference to relevant work suggested both in lines 54-55 and 400-402. Following your suggestion, we also added a remark on the limited practical benefits of marginalization shown by Chrikova et al, 2023 (line 388-389). We will be happy to include Ex 8.1 in the main text of the final version if space permits, as we agree that it can be very helpful for the reader. Finally, we thank you for pointing out the typos, which we have also corrected in the current version.

---

> ### Author Response · Authors · 2024-11-23
>
> We have also remarked that, among the strengths, you mention that “the presentation is excellent”, while the score you gave for the presentation is “3: good”. If you don’t mind us asking, would you agree to update the presentation score (and, if applicable, the final rating) to match the text of your evaluation?

---

> > ### Comment · Reviewer_yPjZ · 2024-11-26
> > **Reviewer Response**
> >
> > I thank the authors for their explanations and especially the addition of the relevant work on linguistic segmentation. I have increased my score.
> >
> > The original score of "3: good" for presentation was reflective of the fact that this category is supposed to include contextualization relative to prior work ("Please assign the paper a numerical rating on the following scale to indicate the quality of the presentation. This should take into account the writing style and clarity, as well as contextualization relative to prior work."), and I had reservations about this point. However, since this has been changed in the updated version of the paper, I have increased this score as well.

---

> > > ### Author Response · Authors · 2024-11-26
> > >
> > > We would like to sincerely thank the reviewer for considering our remarks and adjusting the score.

---

### Meta-Review · Area_Chair_9xuA · 2024-12-23

**Metareview:**

This paper presents a foundational perspective on tokenization and established some formal properties of tokenization; the authors established these properties via the utilization of stochastic maps, which is novel.

Strengths:
- Very clear exposition.
- Novel work where there has been very little theoretical work on tokenization.

Weaknesses:
- No empirical validation of the framework (which is not a huge issue in my opinion).

**Additional Comments On Reviewer Discussion:**

There has been significant discussion between the reviewers and authors where the latter made a significant effort in allaying concerns from reviewers and provided answers.

---

### Decision · Program_Chairs · 2025-01-22

Accept (Poster)